# Circadian Variation of Peripheral Blood Cells in Horses Maintained in Different Environmental and Management Conditions

**DOI:** 10.3390/ani13111865

**Published:** 2023-06-03

**Authors:** Francesca Aragona, Francesca Arfuso, Francesco Fazio, Salvatore De Caro, Elisabetta Giudice, Vincenzo Monteverde, Giuseppe Piccione, Claudia Giannetto

**Affiliations:** 1Department of Veterinary Sciences, University of Messina, Viale Palatucci, 98168 Messina, Italy; francesca.aragona@unime.it (F.A.); farfuso@unime.it (F.A.); ffazio@unime.it (F.F.); egiudice@unime.it (E.G.); gpiccione@unime.it (G.P.); 2Department of Engineering, University of Messina, C/Da Di Dio (S. Agata), 98166 Messina, Italy; salvatore.decaro@unime.it; 3Zooprophylactic Institute of Sicily “A. Mirri”, 90129 Palermo, Italy; vincenzo.monteverde@izssicilia.it

**Keywords:** horse, circadian physiology, season, CBC, environmental changes

## Abstract

**Simple Summary:**

Circadian rhythms promote mammals’ temporal organization as an evolutionary mechanism of adaptation. It is well-known that the physiological status and well-being of domestic animals may be influenced by endogenous and exogenous parameters. The aim of the present study was to evaluate the circadian rhythm of the blood cell count and leukocyte subpopulations in horses maintained in different housing conditions during the four seasons. All hematological parameters and leukocyte cells showed a different trend influenced by housing conditions and seasons. All hematological parameters showed a daily rhythmicity during spring in horses housed in a loose box and paddock. Lymphocytes and neutrophils showed a daily rhythm in horses housed in loose boxes during spring and summer and in paddocks during winter.

**Abstract:**

The aim of our study was to analyze circadian rhythm of the hematological profile of horses housed in a loose box and paddock during the different seasons (spring, summer, autumn, and winter). Blood samples were performed every 4 h for 48 consecutive hours. Red blood cells (RBCs), hemoglobin (HGB), hematocrit (HCT), white blood cells (WBCs), platelets (PLTs), and leukocyte subpopulations (neutrophils, basophils, eosinophils, lymphocytes, and monocytes) were analyzed, and, at the same time, environmental conditions were recorded. A statistically significant effect of housing conditions (*p* < 0.0001) was observed on all hematological values except for WBC during winter and for neutrophils (*p* < 0.0001) during spring and autumn. A statistically significant effect of season (*p* < 0.0001) was found for RBC, HCT, and PLT and for all leukocyte cells (*p* < 0.0001) except for basophils. The single Cosinor method revealed a daily rhythm of hematological parameters during spring in both groups, and a daily rhythm for lymphocytes and neutrophils was observed during spring and summer in horses kept in a loose box and during winter in horses housed in a paddock. Our results revealed that the response of the immune system is regulated by circadian physiology. Knowledge of the periodic temporal structure of mammals should be considered when evaluating animals’ adaptation to temporizations imposed by the environment.

## 1. Introduction

The physiological and psychological ability of the animal to cope with environmental changes is a crucial requirement for the maintenance of homeostasis and animal health and welfare. The inability to cope with environmental pressures can lead individuals to experience stress and potential negative impacts on their health and well-being [1]. Climatic mutation, caused by seasonal changes, represents a stressor factor that may affect animals’ homeostasis [2]. The annual cycle of changing day length (natural photoperiod) provides a reliable environmental cue to determine the time of year. This temporal information is used in domestic animals to guarantee environmental adaptations over the seasons in order to anticipate environmental stressors and, accordingly, ensuring the maintenance of biological functions promoting reproduction and survival [3,4]. Environmental factors are known to act as synchronizers of biological rhythms in mammals, and, in particular, circadian rhythms promote mammals’ temporal organization as an evolutionary mechanism of adaptation [5,6]. This animals’ physiological capability, promoted by the circadian system, is an integral component of homeostasis that interacts with the immune system to guarantee survival [7,8]. The mammalian circadian timing system is composed of many individual clocks, and these are synchronized by a central pacemaker, which coordinates their physiology and behavior following a daily cycle by synchronizing the cell with its environment [5]. Circadian clocks are hierarchically organized, with the central pacemaker located in the (SCN) in the brain and subordinate clocks in almost all peripheral tissues which generate the rhythmic phenomena, including behavior of physiological activities and hematological and hematochemical parameters [8,9]. The circadian rhythm of the hemato-immune system seems to be synchronized by two clocks: the first is exogenous, based on environmental immune stimuli; and the second is endogenous, located in the suprachiasmatic nucleus (SCN), promoting seasonal-change reactions against environmental changes by modulating the dynamic of blood components in horses [10,11,12]. Relative knowledge of hematological profile is an important tool to obtain information about the health status of domestic animals that can be used as an index to monitor physiological and pathological conditions in horses, as it reflects specific changes in animals’ body system [11,13,14]. The hematological profile, including the number of red blood cells, together with the direct erythrocyte’s parameters (hemoglobin and hematocrit), blood coagulation, and immunity parameters, represents an important tool in the determination of physiological changes occurring in animals [11,15]. Thus, the number of circulating leukocytes and the percentage of each leukocyte type show daily rhythmic variations in peripheral blood in humans and mice [16,17,18]. Is well established that immune function is often affected by seasonal changes in domestic animals and regulated by circadian rhythms to set the time of multiple processes controlling the immunological surveillance and response to infection [17,18,19,20]. Therefore, few studies have been carried out concerning the circadian rhythms of hematological parameters and, in particular, concerning leukocyte activity possibly influenced by seasonal changes in horses [21,22,23,24]. The objective of this study was to evaluate the circadian rhythm of the hematological profile and the leukocyte count modifications during the different seasons in horses housed in a loose box and paddock in order to check how seasonal changes and different housing conditions may influence the health status and physiological profile of domestic horses under natural environmental conditions.

## 2. Materials and Methods

### 2.1. Animals

This study was conducted according to the European Directive 2010/63/EU, with current Italian legislation regarding animals’ protection involved for scientific purposes. The protocol of this study was reviewed and approved in accordance with the standards recommended by Ethical Committee of University of Messina (06/2022). All horses were from the same private horse-training center in Sicily (Italy; Latitude 38°7′ N; Longitude 13°22′ E), under a Mediterranean climate, following a weekly show-jumping training program with two days of rest. The training was suspended during the experimental period. A total of 10 clinically healthy Italian Saddle Horses (4 not pregnant and not lactating mares and 6 geldings) that were aged between 10 and 15 years old and with a mean body weight of 445 ± 30 kg were enrolled in this study with the owner consent. Animals were divided into two equal groups maintained under a natural photoperiod. One group, including 2 mares and 3 geldings, was housed in individual loose boxes (3.50 × 3.50 m) in the same stable, without blankets. All individual loose boxes were equipped with a window of 1.50 × 1.50 m, usually manually opened during the day, closed at night on cold periods, and constantly open during warm weather; a grid placed in the front wall of the loose boxes allowed horses to interact. The other group, including 2 mares and 3 geldings, was housed in individual paddocks (500 m/horse with trees to shelter from the sun and rain on a rocky sand-based soil) without blankets. Before the experimental period, each subject underwent a clinical examination (measurement of body temperature, heart rate, respiratory rate, fecal consistency, appetence, and hematological and chemical profile) to monitor its health status and exclude animals with injuries. Each horse was free from internal and external parasites (regularly treated every three months), regularly subjected to Coggins tests (one time a year), and vaccinated against influenza and tetanus. The annual vaccinations were performed three months prior to the experimental period, and the last one was postponed until the end of the experiment [25,26]. All horses were fed three times a day (6:30, 12:00, and 19:00). All animals received the same diet based on a good-quality hay and maintenance concentrates individually. Water was available *ad libitum* in both groups. During the experimental period, thermal and hygrometric records and ventilation were monitored in both groups by using a multiparameter probe (Testo 400). Based on the formula adapted from Thom (1959), the temperature humidity index (THI) was evaluated: THI: (0.8 × ambient temperature) + {[(relative humidity/100) × (ambient temperature—14.4)] + 46.4} [27].

### 2.2. Hematological Analysis

Sampling was performed every 4 h for 48 h, starting from 13:00, at resting period, for each subject at four different times: vernal equinox on 20–21 March 2022 (sunrise recorded at 05:30; sunset recorded at 18:30), summer solstice on 20–21 June 2022 (sunrise recorded at 04:00; sunset recorded at 20:00), autumn equinox on 20–21 September 2022 (sunrise recorded at 05:20; sunset recorded at 18:20), and winter solstice on December 20th–21st 2022 (sunrise recorded at 06:40; sunset recorded at 17:10). Blood samples were collected via a catheter (FEP G14; 13.5 cm) introduced into one of the jugular veins and secured in place with a suture and bandage. An extension tube was attached to the catheter to facilitate blood sampling into vacuum tubes containing ethylenediaminetetraacetic acid (EDTA); immediately, a blood smear was performed from the EDTA tubes. After air-drying, the slides were stained using the the May–Grünwald stain, which consists of successively applying two neutral stains, the May–Grünwald mixture (1902) derived from the Romanowsky mixture (1891) and the Giemsa mixture (1904). On preparations fixed by rapid drying, we highlighted, in particular, the basic or acidic character of the cytoplasm and the granulations of the leukocytes. The microscopic analysis of blood films was performed by using an optical microscope (Nikon Eclipse e200, Nikon Instruments Europe BV, Amsterdam, The Netherlands) at 1000× magnification, with oil. Leukocyte identification and counting were performed on all samples, using a manual 100-cell differential count to identify neutrophils, basophils, eosinophils, lymphocytes, and monocytes on each blood film. All blood samples were refrigerated at 4 °C and analyzed for complete blood count within 2 h. Hematological parameters (red blood cells (RBCs), hemoglobin (HGB), hematocrit (HCT), white blood cells (WBCs), and platelets (PLTs)) were evaluated by using an automated hematology analyzer (HeCo Vet C, SEAC, Florence, Italy).

### 2.3. Statistical Analysis

Before the start of the experimental protocol, inter-subject variabilities were computed as the standard deviations of the means. For each hematological parameter, the standard deviations of the means for each of the 2 days across the seven subjects were applied as the measure of inter-subject variability, excluding endogenous influence [28]. Data were normally distributed (*p* > 0.05, Kolmogorov–Smirnov test) and reported as mean ± standard deviation (SD). An unpaired Student’s *t*-test was applied to investigate possible statistical differences of environmental conditions between the loose box and paddock groups during the experimental period. A General Linear Model (GLM) was applied to the hematological values (RBC, HGB, HCT, WBC, and PLT) and leukocyte cells (neutrophils, basophils, eosinophils, lymphocytes, and monocytes) to evaluate the effect of day, time of day, housing conditions, and seasons. The periodic phenomenon was analytically evaluated by the application of a trigonometric statistical model to each obtained value at each time-series measurement in order to assess the main rhythmic parameters (mesor, amplitude, acrophase, and robustness) by means of the single Cosinor procedure [29]. A Factorial ANOVA was applied to rhythmic parameters to establish the effect of day, time of day, housing conditions, and seasons. Bonferroni’s test was applied for post hoc comparison. A *p*-value < 0.05 was considered to be statistically significant. Data were analyzed using the statistical software STATISTICA 7 (StatSoft Inc., Tulsa, OK, USA).

## 3. Results

During the experimental period, environmental conditions were monitored and the temperature–humidity index was calculated by following the normal seasonal pattern for the Mediterranean area, as shown in Table 1. The mean ambient temperature, THI, and relative humidity did not show statistical differences between groups during the four seasons. The ventilation parameter was statistically higher in horses housed in the paddock than in horses housed in the loose box (*p* < 0.001). All data followed the physiological range of hematological parameters in horses [30]. In the absence of inter-subject variability of the investigated parameters, the application of GLM to the recorded values showed a statistically significant effect of housing conditions (*p* < 0.0001) on all hematological values during winter, except for WBC (*p* = 0.63), and a statistically significant effect of season (*p* < 0.0001) was found for all studied parameters. No statistically significant difference was observed between the time of day and between two days of monitoring for hematological parameters. A statistically significant influence of housing condition (*p* < 0.0001) was found for neutrophils, and a statistically significant effect of season (*p* < 0.0001) was observed for all leukocyte cells, except for basophils (*p* = 0.29). Figure 1 and Figure 2 show the results of Bonferroni’s post hoc comparison. No statistically significant effect for leukocyte cells was observed between the time of day and two days of monitoring. A different daily rhythmic behavior was observed for the studied parameters, as shown in Table 2. A daily rhythm was observed for all hematological parameters during the spring season in the loose-box group and during the winter in the paddock group. Lymphocytes and neutrophils showed a daily rhythm in loose-box horses during the spring and summer and in the paddock group during the winter period. No daily rhythm for hematological values was found during other seasons in both groups. The Factorial ANOVA on hematological parameters and leukocyte population showed no statistical differences between the time of day, two days of monitoring, housing condition, and season on Mesor, amplitude, acrophase, and robustness. 

## 4. Discussion

Seasonal variations influence animals’ physiological response and hematological profile [31,32]. A very similar ambient temperature value between summer and autumn and between spring and winter may be observed on the assumption that the measurements were taken at the beginning of each season (solstices and equinoxes). Since the summer solstice was a quite windy period (Table 1), it is possible for the temperature to remain mild considering the temperature fluctuations between day and night. Moreover, the horses housed in the loose box were not in direct contact with the sun, and the horses maintained in paddocks usually resided in the ventilated shade during the hottest hours.

Hematological parameters varied between horses housed in a loose box vs. horses housed in a paddock. The periodic changes observed may be the result of different factors, such as the influx of some younger formed elements, the distribution between the circulating and the marginal cell compartments, and the distribution between different tissue and organs, which may be themselves rhythmic [33]. Animals accustomed to living according to the Mediterranean temperature will adapt their thermoregulation mechanisms, so group differences may be due to excessively cold/hot conditions to which horses were subjected to, referring to the temperature range to which they were normally exposed to in this area. For horses housed in a paddock, the RBC value was significantly higher in the summer than in the spring, in accordance with Satué et al. [10]. As for the RBC values, HCT was significantly higher in the summer period than during other seasons in horses housed in the loose box, and it was significantly higher in the summer than the winter in the paddock group, confirming the close relation between the HCT value and RBC count, in contrast to Mirzadeh et al.’s (2010) findings [34]. These variations were associated with metabolic acclimation during environmental changes. Higher temperatures occurring during summer may be associated with our results. Higher temperatures activate thermoregulatory mechanism in mammals with a decreasing in body fluids as an adaptive response to heat stress. The increase in the HCT value is associated with the RBC one during summer, thus demonstrating the real increase in both hematological parameters [10]. HGB values reported a statistically significant decrease during winter compared to other seasons in horses housed in a loose box, possibly associated with an increase in the energy and protein requirements during cold weather. Platelets statistically increase their values during the autumn period compared to other seasons in both groups. PLTs are important components of the thrombotic process, and the seasonal variability of stroke and other vascular events could reflect the seasonal variability of platelets, together with a more proinflammatory blood environment in humans [35]. Horses kept in paddocks are naturally and constantly exposed to many thermal stressors, such as the direct solar radiation, ambient temperature, high humidity, and rainfall during the day, and they must alter their behavior and physiology in order to restore homeostasis [31]. Our results showed a statistical seasonality effect in relation to the WBC and its subpopulations. The circannual cycle in cell-mediated immunity has been described in humans and canine species. In particular, circannual variation has been described in the relative number of circulating B and T lymphocytes [33]. The main causes of seasonal variations in hematology may be attributed to climatic changes or day length. All the environmental temperatures recorded were within the thermoneutral zone for horses; the influence of season on WBC could probably be due to the different photoperiod [36]. A statistical increase in lymphocytes during summer compared to other seasons was observed in both groups. These results were probably determined by different environmental changes and influenced by possible subclinical infections. Significant increases in neutrophils’ count were obtained in spring compared to other seasons and for horses housed in a loose box, and they were higher in summer than in winter for the paddock group. Eosinophil values were higher during summer compared to other seasons in the paddock, where external triggering agents for dermatitis, allergies, or parasites are less controlled for compared to the loose box, in accordance with previous studies [37]. The housing system and microclimatic conditions affect the physiological status and different blood parameters in the animal body and have important effects on animals’ circadian rhythms [38]. The environment in which horses are kept influences the ability to keep their thermal status constant, which is related to their thermal characteristic and regulatory physiological mechanism [39,40]. Based on the present results daily rhythm were found for hematological parameters (RBC, HGB, HCT, WBC, and PLT) during the spring season in horses housed in a loose box and paddock with a nocturnal acrophase, as previously demonstrated by other studies [4,6]. Spring is considered to be a milder season compared to the others, in which the ambient temperature more closely reflects the horse’s thermoneutral zone in both housing conditions [5]. Leukocyte subpopulations displayed different rhythmic behaviors. Lymphocytes and neutrophils showed a circadian diurnal rhythm during spring and summer for horses housed in a loose box, as previously demonstrated by Shina et al. (2019) but in contrast with other findings in which circulating leukocytes showed a nocturnal rhythm in bovines, mice, and humans, justifying these differences to species factor, different blood sampling period, or different management conditions [2,23,41]. No daily rhythm was observed for monocytes, basophils, and eosinophils during the four seasons in loose box and paddock. Daily changes in blood leukocyte counts have been attributed to a rhythmic cell distribution between the peripherical tissues and circulating blood compartments and to a rhythmic influence of new cells [2]. Although all leukocyte populations contribute to the circadian rhythm, our results showed that lymphocytes and neutrophils were the main characters to that oscillation. Lymphocytes’ circadian rhythm showed a diurnal acrophase during early morning in spring that is delayed to the afternoon during summer for horses housed in a loose box. A similar behavior was observed for a neutrophil population expressing a diurnal acrophase during the early morning in the spring season and a diurnal rhythmicity with an acrophase during the afternoon in the summer, with a high robustness of rhythm. This rhythmic distribution reflects the immune system physiology during the active phase of the horse, where it is the most probable for the antigen to enter, requiring an important energy expenditure [13]. Therefore, lymphocytes migrate from blood to lymphoid tissues when the entering of the antigen is most likely to occur. Accordingly, during the resting period, the number of leukocyte cells has minimal values [3,23]. During the early morning, the autonomous nervous system and the neuroendocrine system have been shown to modulate leukocyte physiology, supporting the concept that circadian timing is an important aspect of hypothalamic–immune communication in humans [42]. A daily rhythm of lymphocytes and neutrophils was observed for horses housed in a paddock during the winter season, confirming that seasons and different environmental parameters may influence the hematological rhythm in horses. While excessive heat causes the disruption of the rhythm, a cold environment stabilizes it but does not affect the total count, as the cells are already balanced, as in our case.

## 5. Conclusions

In light of our results, we can conclude that hematological parameters show a different circadian rhythmic behavior in horses housed in a loose box and horses housed in a paddock during the four seasons. 

This finding contributes to the knowledge about the impact of the management conditions on the physiological status of horses. During the four seasons, a different response of the immune system is regulated by the circadian physiology, influencing the horse’s well-being. This study offers chronobiological support to understanding the adaptation of horses to temporizations imposed by the environment. The knowledge of the periodic temporal structure of horses makes it possible to understand its functions, being very useful in the management and prevention diagnosis, as well as allowing us to detect the consequences that environmental changes may have on temporal organization.

## Figures and Tables

**Figure 1 animals-13-01865-f001:**
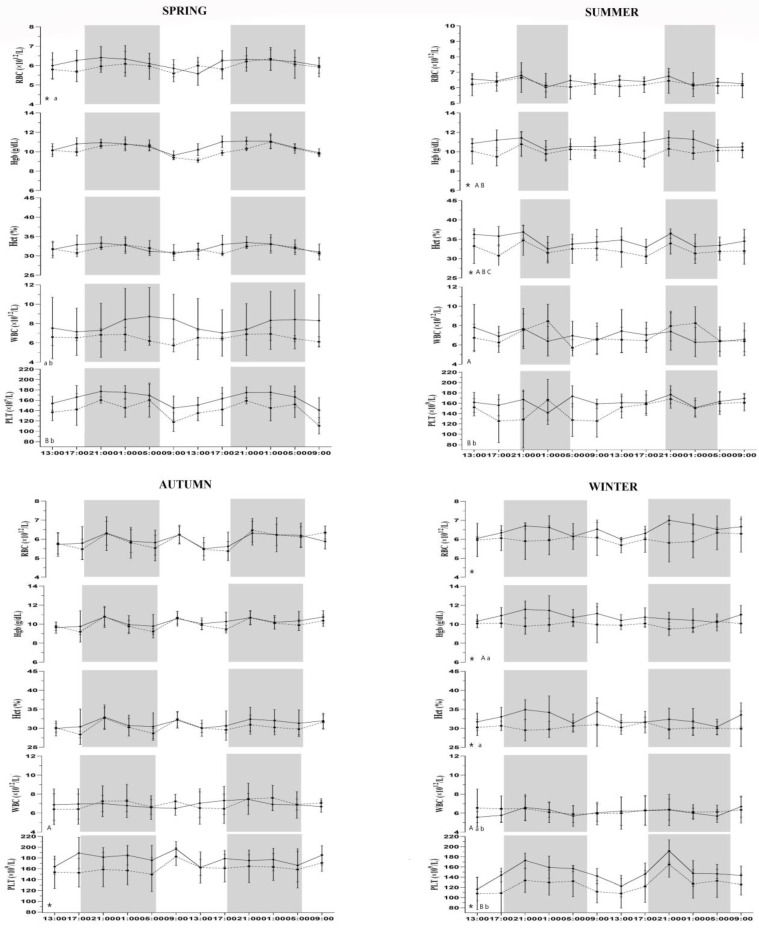
Trend of hematological parameters for horses housed in loose box (full line) and horses housed in paddock (dotted line) expressed in their conventional units recorded every 4 h for a period of 48 h during spring, summer, autumn, and winter. Each dataset corresponds to the mean ± standard deviation (SD) of our results. Statistical significances: Capital letters indicate the statistical differences between seasons in loose boxes (A vs. spring; B vs. autumn and C vs. winter). Lowercase letters indicate the statistical differences between seasons in paddock (a vs. summer; b vs. autumn). Symbols (*) indicate statistical differences between groups. Gray and white areas represent the scoto- and photophase of experimental period for each season.

**Figure 2 animals-13-01865-f002:**
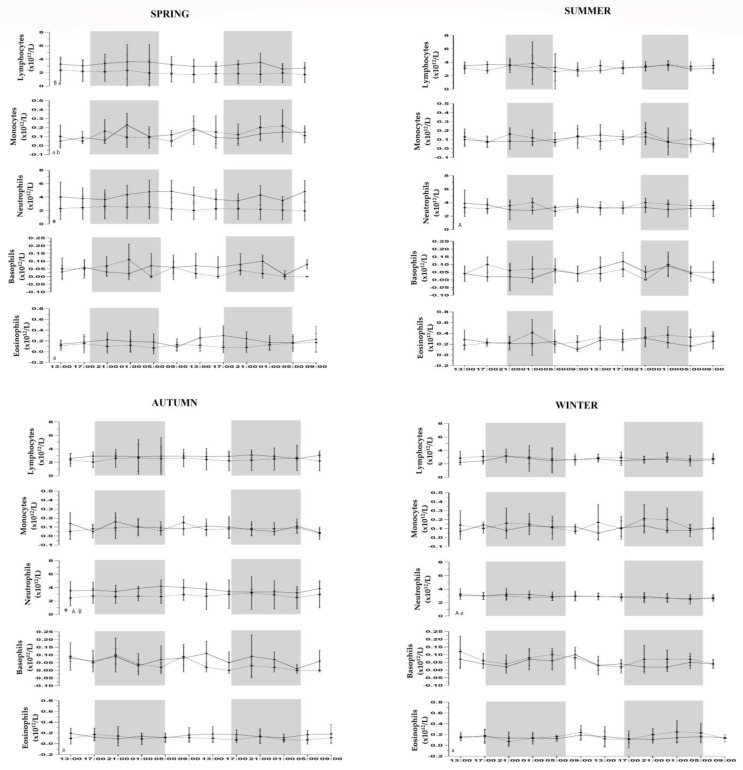
Trend of hematological parameters for horses housed in loose box (full line) and horses housed in paddock (dotted line), expressed in their conventional units recorded every 4 h for a period of 48 h during spring, summer, autumn, and winter. Each dataset corresponds to the mean ± standard deviation (SD) of our results. Statistical significances: Capital letters indicate the statistical differences between seasons in loose boxes (A vs. spring; B vs. summer and vs. autumn). Lowercase letters indicate the statistical differences between seasons in paddock (a vs. summer; b vs. winter). Symbols (*) indicate statistical differences between groups. Gray and white areas represent the scoto- and photophase of experimental period for each season.

**Table 1 animals-13-01865-t001:** Mean values ± SD of ambient temperature, relative humidity, ventilation, and temperature–humidity index (THI) expressed in their conventional unit, together with their statistical significances recorded during each season for box and paddock groups. Lowercase letters indicate the statistical differences between groups.

	BOX	PADDOCK
SPRING		
Ambient temperature (°C)	16.40 ± 2.26	16.47 ± 0.39
Relative humidity (%)	61.00 ± 0	44.76 ± 0
Ventilation (m/s)	0.03 ± 0 ^a^	0.28 ± 0.08 ^a^
Temperature–humidity index (THI)	60.74 ± 2.47	60.50 ± 3.32
SUMMER		
Ambient temperature (°C)	25 ± 0	23.50 ± 0
Relative humidity (%)	63 ± 0	61.50 ± 0.01
Ventilation (m/s)	0.4 ± 0.01 ^a^	1.38 ± 0.34 ^a^
Temperature–humidity index (THI)	73.08 ± 2.79	70.75 ± 4.09
AUTUMN		
Ambient temperature (°C)	23.98 ± 0.96	23.15 ± 1.88
Relative humidity (%)	75.00 ± 6.57	73.83 ± 6.50
Ventilation (m/s)	0.00 ± 0.00 ^a^	0.52 ± 0.68 ^a^
Temperature–humidity index (THI)	72.76 ± 1.31	71.39 ± 1.60
WINTER		
Ambient temperature (°C)	16.33 ± 1.74	15.00 ± 1.02
Relative humidity (%)	83.00 ± 9.49	82.50 ± 8.19
Ventilation (m/s)	0.00 ± 0.00 ^a^	0.17 ± 0.14 ^a^
Temperature–humidity index (THI)	61.06 ± 1.73	58.89 ± 1.90

**Table 2 animals-13-01865-t002:** Mean values ± DS of circadian parameters (mesor, amplitude, acrophase, and robustness) expressed in their conventional unit for all leukocyte subpopulations investigated (lymphocytes, monocytes, neutrophiles, eosinophils, and basophils) during the four seasons in horses housed in loose box and paddock.

	Box				Paddock			
	Mesor (1012/L)	Amplitude (%)	Acrophase (hh:mm)	Robustness (%)	Mesor (1012/L)	Amplitude (%)	Acrophase (hh:mm)	Robustness (%)
SPRING								
LYMPHOCYTES	3.24 ± 1.30	0.68 ± 0.32	6:15 ± 0.26	87.05 ± 13.21	No Rhythmicity			
MONOCYTES	No Rhythmicity							
NEUTROPHILS	4.13 ± 1.41	1.05 ± 0.62	06:27 ± 0.33	77.08 ± 10.31				
EOSINOPHILS	No Rhythmicity							
BASOPHILES	No Rhythmicity							
SUMMER								
LYMPHOCYTES	2.99 ± 0.08	0.59 ± 0.21	18:11 ± 0.02	92.1 ± 7.55	No Rhythmicity			
MONOCYTES	No Rhythmicity							
NEUTROPHILS	2.04 ± 0.12	0.43 ± 0.019	17:50 ± 0.52	93.9 ± 9.12				
EOSINOPHILS	No Rhythmicity							
BASOPHILES	No Rhythmicity							
AUTUMN								
LYMPHOCYTES	No Rhythmicity				No Rhythmicity			
MONOCYTES								
NEUTROPHILS								
EOSINOPHILS								
BASOPHILES								
WINTER								
LYMPHOCYTES	No Rhythmicity				2.98 ± 0.96	0.51 ± 0.31	20.20 ± 0.05	84.17 ± 15.75
MONOCYTES					No Rhythmicity			
NEUTROPHILS					2.99 ± 0.76	0.51 ± 0.43	19.27 ± 0.17	96.8 ± 17.43
EOSINOPHILS					No Rhythmicity			
BASOPHILES					No Rhythmicity			

## Data Availability

The datasets used and/or analyzed during the current study are available from the corresponding author upon reasonable request.

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
