# Peer review of "Circadian Variation of Peripheral Blood Cells in Horses Maintained in Different Environmental and Management Conditions"

_animals, 2023, doi:10.3390/ani13111865_

Round 1

Reviewer 1 Report

Dear Editor, the study entitled "Comparative evaluation of the circadian oscillation during different season in numbers peripheral blood cells of the horses maintained in different management conditions"showed interesting findings related to the effects of circadian rhythm on the response of immune system of horse. The study emphasizes the evidence that immune system follows a circadian regulation and highlights the usefulness of periodic temporal structure of mammals as a support to understand animals’ adaptation to temporizations imposed by the environment.

The information is of significant interest to the Journal's readers, and the topic of the manuscript is proper for the Journal. However, some minor changes could improve the manuscript.

I suggest that the study is suitable for publication after minor changes

Specific comments

I suggest to improve the title as it could be misinterpreted by the lector in the current form, and, to my opinion, it does not reflect the study appropriately. Authors could write “Circadian oscillation of peripheral blood cells in horses maintained in different environmental and management conditions”

Please change “leukocytes subpopulation” with “leukocyte subpopulations” in summary and throughout the manuscript.

Line 13: please remove “and for this reason”

Line 14-16: please reduce the paragraph by report “blood cell count”

Line 17: please delete “(box and paddock)”

Lines 18-21: Please change “All hematological parameters and leukocyte cells showed a different trend influenced by housing conditions and seasons.All hematological parameters showed a daily rhythmicity during spring season in horses housed in box and paddock. Lymphocytes and neutrophils showed a daily rhythm in horses housed in boxes during spring and summer and in paddocks during winter.”

Please, change “…in box and paddock…” as “…in horses housed in box and paddock…” throughout the text.

Please make uniform “hematology/haematological parameters” or “Haematology/haematological parameter” throughout the text.

Please, delete the last sentence of the simple summary and moved it at the end of the abstract.

More information related to animals enrolled in the study should be added in the abstract section (i.e. the number of enrolled animals, gender).

In the sentence (Lines 29-30) “Red blood cells- RBC, hemoglobin- HGB, hematocrit- HCT, white blood cells- WBC, platelets- PLT and leukocytes counts (neutrophils, basophils, eosinophils, lymphocytes and monocytes)” WBC is the same of leukocyte count! Please change “…and leukocytes counts (neutrophils, basophils, eosinophils, lymphocytes and monocytes)…” as “…and leukocyte subpopulations (neutrophils, basophils, eosinophils, lymphocytes and monocytes).”

Lines 31-36: I suggest to clarify the paragraph.

Introduction sound well and it falls within the main topic of the study. Authors cited appropriate references throughout the section.

I suggest to rewrite the first  sentence of introduction “Animal welfare represent the physiological and psychological ability of the animal to cope with the environmental changes in order to guarantee their well-being.”, since it could be misinterpreted by the lector. Authors could write “The physiological and psychological ability of the animal to cope with the environmental changes is a crucial requirement for the maintenance of homeostasisand to ensure the animal health status and welfare.”

Lines 46-47: Please check and improve the sentence “Climatic mutation, caused by seasonalchanges, represents a stressor factor that may affects animals’ [2].” I think that the final word is missing.

Line 65: please delete “suprachiasmatic nucleus”

Lines 70-75: Please improve the paragraph “Hematological profile including number of red blood cells (RBC), involved in oxygen transportation, hemoglobin concentration (HGB), hematocrit (HCT), blood coagulation (PLT) and immunity represented by total number of white blood cells (WBC) and their different types (Neutrophils, basophils, eosinophils, lymphocytes and monocytes) is important inthe determination of physiological changes occurring in animals [11, 15].” Authors could change it as “Hematological profile including number of red blood cells together with the direct erythrocytes parameters (hemoglobinand haematocrit), blood coagulation and immunity parameters represent important tool in the determination of physiological changes occurring in animals [11, 15].”

The section of Materials and Methods is clear for the reader and well describes the methods applied in the study.

Line 98,Please put “including 2 mares and 3 geldings” after one group.

Line 99, please put “including 2 mares and 3 geldings” after the other group.

Lines 112-115: please move the sentence “All horses were taken from the same private horse training center in Sicily (Italy; Latitude 38° 7’ N; Longitude 13° 22’ E) under a Mediterranean climate, following a weekly show jumping training program with two days of rest. The training was suspended during the experimental period.” in animal subsection.

Line 121: please check “[2.09105 mm])”

Line 174: please check the sentence “Figure 1-8 showed the results of Bonferroni’s post hoc comparison.”

Please, in each figure legend correct “in box and in paddock” as “in horses housed in box and in paddock”

Results section as well as Discussion section is clear and well written and the findings obtained were properly discussed. However, please, carefully check the discussion section and improve English language and punctuation (an example: Line 321,Platelets statistically increase its values during autumn…” should be “Platelets statistically increase their values during autumn…” or “Platelet count statistically increase its values during autumn…”).

Please make uniform “hematology/haematological parameters” or “Haematology/haematological parameter”

Conclusion section well summarizes the main findings of the study and it well emphasizes the significance of the study. However, English language should be improved. Authors shul rewrite the first sentence of the section “In the light of what has been obtained, we can say that the various components of the… showed the same photoperiod but different temperatures.”

Overall, the paper is well written and, since it adds useful information about comparative evaluation of the circadian oscillation during different season in numbers peripheral blood cells of the horses, it deserves to be published after minor revisions

Author Response

Dear Reviewer,

thank you very much for the review of the Manuscript entitled “Comparative evaluation of the circadian variation during different seasons in hematological profile of the horses maintained in different management conditions

We were glad to read that reviewer recommended the work for publication. We have been able to address the majority of the reviewers concerns as we have outlined in detail below and we have modified the manuscript accordingly.

We hope that the modified manuscript will now be acceptable for publication.

Thanks you for your help and suggestions. Looking forward to your comments on our final version.

Yours sincerely,

Claudia Giannetto

---------------------------------------------------------------------------------------

I suggest to improve the title as it could be misinterpreted by the lector in the current form, and, to my opinion, it does not reflect the study appropriately. Authors could write “Circadian oscillation of peripheral blood cells in horses maintained in different environmental and management conditions”

The title has been changed in: “Circadian variation of peripheral blood cells in horses maintained in different environmental and management conditions” according to the suggestions.

Please change “leukocytes subpopulation” with “leukocyte subpopulations” in summary and throughout the manuscript.

“leukocyte subpopulations” has been corrected along the text

Line 13: please remove “and for this reason”

“and for this reason” has been removed

Line 14-16: please reduce the paragraph by report “blood cell count”

The paragraph has been reduced reporting “blood cell count”

Line 17: please delete “(box and paddock)”

“(box and paddock)” has been deleted

Lines 18-21: Please change “All hematological parameters and leukocyte cells showed a different trend influenced by housing conditions and seasons.All hematological parameters showed a daily rhythmicity during spring season in horses housed in box and paddock. Lymphocytes and neutrophils showed a daily rhythm in horses housed in boxes during spring and summer and in paddocks during winter.”

The section has been changed as suggested

Please, change “…in box and paddock…” as “…in horses housed in box and paddock…” throughout the text.

“…in horses housed in box and paddock…” has been changed throughout the text

Please make uniform “hematology/haematological parameters” or “Haematology/haematological parameter” throughout the text.

“hematology/haematological parameters” has been uniformed throughout the text

Please, delete the last sentence of the simple summary and moved it at the end of the abstract.

More information related to animals enrolled in the study should be added in the abstract section (i.e. the number of enrolled animals, gender).

The last sentence of the summary: “. Our results revealed that the response of immune system is regulated by circadian physiology and the knowledge of periodic temporal structure of mammals should be a support to well un-derstand animals’ adaptation to temporizations imposed by the environment.” Has been moved to the abstract

In the sentence (Lines 29-30) “Red blood cells- RBC, hemoglobin- HGB, hematocrit- HCT, white blood cells- WBC, platelets- PLT and leukocytes counts (neutrophils, basophils, eosinophils, lymphocytes and monocytes)” WBC is the same of leukocyte count! Please change “…and leukocytes counts (neutrophils, basophils, eosinophils, lymphocytes and monocytes)…” as “…and leukocyte subpopulations (neutrophils, basophils, eosinophils, lymphocytes and monocytes).”

“…and leukocyte subpopulations (neutrophils, basophils, eosinophils, lymphocytes and monocytes).” Has been corrected

Lines 31-36: I suggest to clarify the paragraph.

“A statistically significant effect of housing conditions (p<0.0001) was observed on all hematolog-ical values except for WBC during winter and for neutrophils (p<0.0001) during spring and au-tumn. A statistically significant effect of season (p<0.0001) was found for RBC, HCT and PLT and for all leukocyte cells (p<0.0001) except for basophils.” The sentence has been modified

Introduction sound well and it falls within the main topic of the study. Authors cited appropriate references throughout the section.

I suggest to rewrite the first sentence of introduction “Animal welfare represent the physiological and psychological ability of the animal to cope with the environmental changes in order to guarantee their well-being.”, since it could be misinterpreted by the lector. Authors could write “The physiological and psychological ability of the animal to cope with the environmental changes is a crucial requirement for the maintenance of homeostasisand to ensure the animal health status and welfare.”

The sentence has been modified in: “The physiological and psychological ability of the animal to cope with the environmental changes is a crucial requirement for the maintenance of homeostasisand to ensure the animal health status and welfare” as suggested

Lines 46-47: Please check and improve the sentence “Climatic mutation, caused by seasonalchanges, represents a stressor factor that may affects animals’ [2].” I think that the final word is missing.

The final missing word has been rewritten: “homeostasis”

Line 65: please delete “suprachiasmatic nucleus”

“Suprachiasmatic nucleus” has been delated

Lines 70-75: Please improve the paragraph “Hematological profile including number of red blood cells (RBC), involved in oxygen transportation, hemoglobin concentration (HGB), hematocrit (HCT), blood coagulation (PLT) and immunity represented by total number of white blood cells (WBC) and their different types (Neutrophils, basophils, eosinophils, lymphocytes and monocytes) is important inthe determination of physiological changes occurring in animals [11, 15].” Authors could change it as “Hematological profile including number of red blood cells together with the direct erythrocytes parameters (hemoglobinand haematocrit), blood coagulation and immunity parameters represent important tool in the determination of physiological changes occurring in animals [11, 15].”

The sentence has been modified in: “Hematological profile including number of red blood cells together with the direct erythrocytes parameters (hemoglobinand haematocrit), blood coagulation and immunity parameters represent important tool in the determination of physiological changes occurring in animals [11, 15].”

The section of Materials and Methods is clear for the reader and well describes the methods applied in the study.

Line 98,Please put “including 2 mares and 3 geldings” after one group.

“Including 2 mares and 3 geldings” was moved after “one group”

Line 99, please put “including 2 mares and 3 geldings” after the other group.

“Including 2 mares and 3 geldings” was moved after “other group”

Lines 112-115: please move the sentence “All horses were taken from the same private horse training center in Sicily (Italy; Latitude 38° 7’ N; Longitude 13° 22’ E) under a Mediterranean climate, following a weekly show jumping training program with two days of rest. The training was suspended during the experimental period.” in animal subsection.

The section has been moved to the animals subsection as suggested

Line 121: please check “[2.09105 mm])”

Catheter dimension has been modified: “(FEP G14; 13.5 cm)”

Line 174: please check the sentence “Figure 1-8 showed the results of Bonferroni’s post hoc comparison.”

“Figure 1-2 show the results of Bonferroni’s post hoc comparison.” The sentence has been revised

Please, in each figure legend correct “in box and in paddock” as “in horses housed in box and in paddock”

The figure legends the sentence has been improved in: “in horses housed in box and in paddock”

Results section as well as Discussion section is clear and well written and the findings obtained were properly discussed. However, please, carefully check the discussion section and improve English language and punctuation (an example: Line 321,“Platelets statistically increase its values during autumn…” should be “Platelets statistically increase their values during autumn…” or “Platelet count statistically increase its values during autumn…”).

The punctuation has been revised

Please make uniform “hematology/haematological parameters” or “Haematology/haematological parameter”

“hematology/haematological parameters” has been unified throughout the text

Conclusion section well summarizes the main findings of the study and it well emphasizes the significance of the study. However, English language should be improved. Authors shul rewrite the first sentence of the section “In the light of what has been obtained, we can say that the various components of the… showed the same photoperiod but different temperatures.”

The first sentence has been revised: “In the light of what has been obtained, we can say that the various components of the hematological profile showed the same photoperiod but different temperatures during seasons.”

Reviewer 2 Report

Very interesting study which, however, still raises some questions that need to be clarified.

Titel

I would replace oscillation with fluctuation or variation.

Abstract

Line 31 to 36: Why are there two separate sentences of significant effects by housing and season, with different results?

Keywords

I would replace leukocytes by CBC.

Materials and Methods

Line 90 to 100: Please add more information about the keeping of the horses. What kind of ground was used? Did the animals have contact to other horses in the boxes? How were the lighting conditions in the boxes? Were the horses covered in winter? Could the animals in the boxes look outside with their heads?

Line 97-100: It is also the question whether keeping them individual (boxes) had an impact compared to keeping them in groups (paddock)?

Line 98 to 100: This does not match with the number of animals in line 94, now there are 4 mares and 6 geldings.

Line 102: Only CBC or clinical chemistry too? And only at the start of the study?

Line 103: How regularly were parasites checked or treated?

Line 104: Which vaccines and in what time frame prior to study/sampling?

Line 105: How did the animals in the paddock get their maintenance concentrates, individually?

Line 119 to 120: This sentence is a repetition of the sentence in line 115 to 116.

Line 123: Blood smear with or without EDTA?

Line 131: With which magnification? 1000x with oil?

Line 132: lymphocytes

Line 133: After what time were the samples measured?

Results

Line 172: This is with inter-subject variability, right?

Line 174: p-value for basophiles?

Line 179: In the paddock group in winter not in spring!

Line 180: Please move this sentence in line 179.

Line 182 to 183: Could be deleted, repetition of the sentence in line 179.

Figures

Why did you choose 1 p.m. as the starting point and not 9 a.m.? Must also be supplemented in the material and methods section line 115.

Is it possible to connect figure 1 and 5 and so on to reduce the number of figures and make it more manageable?

Discussion

Line 294: And also between spring and winter.

Line 300: What is the exact explanation that the RBC for example is lower in winter and summer in the paddock, there are completely opposite environmental influences?

Line 329 to 330: The temperature differences over the year were very small and especially in winter and spring we are in the range of the preferred temperature of the horses and far away from cold temperatures related to horses. Therefore, the question is, can these temperature differences really have had such a great influence or are the results over-interpreted due to the small number of animals?

Line 331: Please add reference.

Line 377: Actually, keeping in a paddock is preferred because it comes closest to the natural living conditions of the horses and should stress them little compared to the close solitary confinement in boxes with often poor light and air conditions (dust especially in winter).

Minor editing by a native speaker would be advised.

Author Response

Dear Reviewer,

thank you very much for the review of the Manuscript entitled “Comparative evaluation of the circadian variation during different seasons in hematological profile of the horses maintained in different management conditions

We were glad to read that reviewer recommended the work for publication. We have been able to address the majority of the reviewers concerns as we have outlined in detail below and we have modified the manuscript accordingly.

We hope that the modified manuscript will now be acceptable for publication.

Thanks you for your help and suggestions. Looking forward to your comments on our final version.

Best Regards

Claudia Giannetto

   ----------------------------------------------------------------------------------

I would replace oscillation with fluctuation or variation

Oscillation has been replaced with variation

Line 31 to 36: Why are there two separate sentences of significant effects by housing and season, with different results?

The sentences have been unified

I would replace leukocytes by CBC.

Leukocytes has been replaced with CBC

Materials and Methods

Line 90 to 100: Please add more information about the keeping of the horses. What kind of ground was used? Did the animals have contact to other horses in the boxes? How were the lighting conditions in the boxes? Were the horses covered in winter? Could the animals in the boxes look outside with their heads?

other information has been reported

Line 97-100: It is also the question whether keeping them individual (boxes) had an impact compared to keeping them in groups (paddock)?

“All individual boxes were equipped with a window of 1.50 x1.50 m, usually manually opened during the day and closed at night on cold periods, and constantly opened during warm; a grid placed in the front wall of the boxes allowed horses interaction.” This information has been added to clarify their interaction in boxes and paddock each other

Line 98 to 100: This does not match with the number of animals in line 94, now there are 4 mares and 6 geldings.

The error has been corrected in line 94

Line 102: Only CBC or clinical chemistry too? And only at the start of the study?

“biochemical analysis has been added to the section” As said only before the start of the experimental period they underwent the clinically examination for excluding animal with possible injuries to the entire study “…and exclude animals with injuries.”

Line 103: How regularly were parasites checked or treated?

“Regularly treated every three months” has been included

Line 104: Which vaccines and in what time frame prior to study/sampling?

“Each horse was free from internal and external parasites (regularly treated every three months), regularly subjected to Coggins tests (one time a year) and vaccinated against influenza and tetanus. The annual vaccinations were performed three months prior to the experimental period and postponed the last until the end of the experiment [25, 26].” This information has been included

Line 105: How did the animals in the paddock get their maintenance concentrates, individually?

“individually” has been added

Line 119 to 120: This sentence is a repetition of the sentence in line 115 to 116.

The repeated sentence has been delated

Line 123: Blood smear with or without EDTA?

“from the EDTA” tubes was included

Line 131: With which magnification? 1000x with oil?

“at 1000x magnification with oil” was added to the section

Line 132: lymphocytes

“lymphocytes” has been corrected

Line 133: After what time were the samples measured?

“within 2 hours” has been included

Results

Line 172: This is with inter-subject variability, right?

Yes, as mentioned: “In the absence of inter-subject variability..”

Line 174: p-value for basophiles?

“(p=0.29)” was included

Line 179: In the paddock group in winter not in spring!

“during winter” was corrected

Line 180: Please move this sentence in line 179

“Lymphocytes and neutrophils showed a daily rhythm in box horses during spring and summer season and in paddock group during winter period.” This sentence has been moved

Line 182 to 183: Could be deleted, repetition of the sentence in line 179.

This sentence has been delated

Figures

Why did you choose 1 p.m. as the starting point and not 9 a.m.? Must also be supplemented in the material and methods section line 115.

The starting time was linked to the management of the equestrian centre.

This information has been supplemented: “starting from 13:00”

Is it possible to connect figure 1 and 5 and so on to reduce the number of figures and make it more manageable?

The figures have been connected into two figures

Discussion

Line 294: And also between spring and winter.

and between spring and winter has been added

Line 300: What is the exact explanation that the RBC for example is lower in winter and summer in the paddock, there are completely opposite environmental influences?

The sentence has been changed in “Hematological parameters showed some statically variants between in horses housed in box than horses housed in paddock the periodic changes in the number of circulating in peripheral blood may be the result of different factors, such as, the influx of some young formed elements, the distribution between the circulating and the marginal cell compartments and the distribution between different tissue and organs, which may be themselves rhythmic [33].”

Line 329 to 330: The temperature differences over the year were very small and especially in winter and spring we are in the range of the preferred temperature of the horses and far away from cold temperatures related to horses. Therefore, the question is, can these temperature differences really have had such a great influence or are the results over-interpreted due to the small number of animals?

The sentence “Circannual cycle in cell-mediated immunity has been described in humans and canine species. In particular, circannual variation has been described in the relative number of circulating B and T lymphocytes [33].” Has been added.

Line 331: Please add reference.

“Morgan, K. Thermoneutral zone and critical temperatures of horses. J Therm Biol 1998; 23, 59-61. https://doi.org/10.1016/S0306-4565(97)00047-8” has been added

Line 377: Actually, keeping in a paddock is preferred because it comes closest to the natural living conditions of the horses and should stress them little compared to the close solitary confinement in boxes with often poor light and air conditions (dust especially in winter)

The sentence “that are major subjected to infection and disease during winter period in paddock compared to box where environmental stressors were limited.” Has been deleted

Round 2

Reviewer 2 Report

Line 329: The sentence is still included in the manuscript and has not been deleted.

Author Response

The sentence has been succesfully delated